# Prospect of LNG as Marine Fuel in Indonesia: An Economic Review for a Case Study of 600 TEU Container Vessel

**Riko Butarbutar, Raja Oloan Saut Gurning * and Semin**

Department of Marine Engineering, Faculty of Marine Technology, Institut Teknologi Sepuluh Nopember, Surabaya 60111, Indonesia
* Correspondence: sautg@its.ac.id; Tel.: +62-(0)82140060563

**Abstract:** The alternative use of environmentally friendly marine fuel by Indonesian vessel owners complies with IMO regulations. Marine fuels with low carbon and sulfur are alternative fuels to the current fossil fuels used by the shipping industry. Some alternative marine fuels are being used or developed such as LNG, hydrogen, and methanol. LNG is one alternative fuel that is used significantly as a marine fuel in the shipping industry. As one of the LNG producers, Indonesia is still behind in using LNG as an alternative marine fuel. One of the main reasons is the use of conventional marine fuels such as HFO, MDO, MGO and the understanding of LNG as an expensive and high-risk commodity. However, vessel owners face various challenges when selecting alternative fuel, which is associated with price and technology. This study aims to analyze a 600 TEU container vessel by calculating its net present value, the capital recovery factor and life cycle analysis (LCA) to determine whether owners carry out the investment. The result of the economic analysis for the 600 TEU vessel showed that the investment of retrofit for LNG as a marine fuel will be a good choice for owners due to the challenge of capital cost for financing a new vessel.

**Keywords:** fuel gas supply system; life cycle analysis; LNG

## 1. Introduction

The shipping sector is an important player in the Indonesian economy because sea transportation is cost-effective. Its growth is impacted by indigenous and international regulatory bodies such as IMO. However, the current regulatory standard adopted by IMO is emission control from the vessel's exhaust. Ref. [1] Arefin et al. stated that the increased demand for energy triggers the production of greenhouse gases (GHGs) in enormous quantities. GHGs are obtained from burning fossil fuels, which ultimately cause global warming. Since the implementation of emission control by IMO, several studies have been carried out on alternative fuels, and presently, various types are available in the market. Vessel operators have no choice but to select advanced alternative fuel technology as a management strategy. In terms of sulfur emission, traditional marine fuel is influenced by component and hydrocarbon composition and the structure of asphaltenes [2]. The characteristics, both physical and chemical, of asphaltenes will also impact the sulfur content [3]. The major alternative marine fuels in development are hydrogen, LNG, methanol and batteries [4]. Jack Sharples stated that transportation modes are significant sources of carbon emission (CO2) [5]. Air pollution containing SO, NOx and particulate emissions significantly impacts human health. In 2015, approximately 32.3 billion tons of CO2 emissions were recorded globally, of which 7.7 billion was obtained from the transportation sector with 5.8 billion tons on land. This is followed by sea transportation and aviation, with approximately 657 million tons and 530 million tons of emissions, respectively. While land transportation contributes a huge amount of CO2, NOx, and particulate emissions, the sea contributes approximately 90% of SOx emissions and impacts the local port [5]. One factor that causes high emissions from vessels is the cheap price and filtering technology of fuel.

The use of LNG as alternative marine fuel for decarbonization is implemented in various types of vessels. In South Korea, there was a study on LNG fuel application to new bulk carriers [6]. LNG as a marine fuel is not only implemented in deep-sea shipping, but it has implemented for short-sea shipping or domestic shipping. Fishing vessels are another type of vessel that increase the impact on the environment due to vessel emission, and these types of vessels have started to use alternative marine fuels. The study shows that LNG fuel is a good option for fishing vessels to reduce environmental impact [7]. Another type of vessel that started to use LNG as a marine fuel is the Ro-ro ferry vessel. The conversion of Ro-ro vessels to LNG-fueled vessels will be technically feasible and a good option for local ship operators [8]. In Indonesia, vessel owners are usually more comfortable with conventional fuels such as HFO, MDO or MGO rather than engaging in environmentally friendly vessel operations. According to one study, heavy fossil hydrocarbons are transformed into natural gas, and their constituents can reduce emissions [9]. This flexibility is significant to the termination of dependency on conventional fuel; many countries are yet to develop renewable energy [10]. Yun et al. stated that energy derived from fossil fuels is expensive, impacts the economy, social life protection, welfare, and the educational sector and triggers air pollution [11]. The government needs to create awareness of the importance of environmentally friendly fuel, specifically in the shipping sector. Its realization is bound to impact the economy and social environment for decades significantly. Vessel owners can utilize several fuel alternatives to comply with the IMO regulation. Natural gas is the preferred fuel in this sector due to its innumerable advantages. These include GHGs reduction, better combustion efficiency, attractive cost, and renewability through biomass production [12]. Irrespective of the fact that natural gas is majorly used in the transportation sector due to its availability and environmental benignity, it is still limited to small engines, specifically spark-ignition (SI) and is rarely found in large diesel engines [12]. In the next decade, the number of LNG-fueled vessels is forecasted to increase immensely, even though certain segments are bound to experience massive expansion [13]. The possibility of vessels switching from using fossil fuel to LNG is because this has gained significant concern among the currently evaluated technologies [14]. In this present study, LNG is used as an alternative fuel due to its numerous advantages. These include advanced LNG vessel technology and the market price established across major ports globally. Another journal has studied the LNG fuel and diesel engine based on the Energy Storage System (ESS) using the NGSA-II algorithm and discovered the optimal scheme to reduce pollutants and cost [15].

Previous studies stated that LNG is the most advanced energy technology implemented on board vessels. Its source is readily available in terms of emission reduction compared to other alternative fuels. Natural gas is a promising alternative fuel source in transportation because of its remarkable advantages [1]. A survey was conducted in some shipping companies that render several services related to the container, offshore, general cargo vessels, and crew boats. The objective of the survey was to understand the emission control requirement mandated by IMO and vessel owners' plans for using environmentally friendly marine fuel. Eight companies were selected randomly as respondents, and due to confidentiality, the company name is classified. Their responses were used as sampling representatives with respect to marine fuel transition. These eight companies represented the types of vessels operating in Indonesia such as container vessels, general cargo vessels, oil tanker vessels, LCT and offshore support vessels.

In accordance with the distributed questionnaires, most respondents (70.6%) stated that they knew about IMO regulation to reduce sulfur content and even felt the impact on their businesses. The graphic representation in Figure 1 shows that the rest of the respondents do not understand the IMO regulation on emissions. The challenge is to ensure vessel owners in the country realize that IMO regulation should be implemented in target regions to obtain zero emissions by 2050.

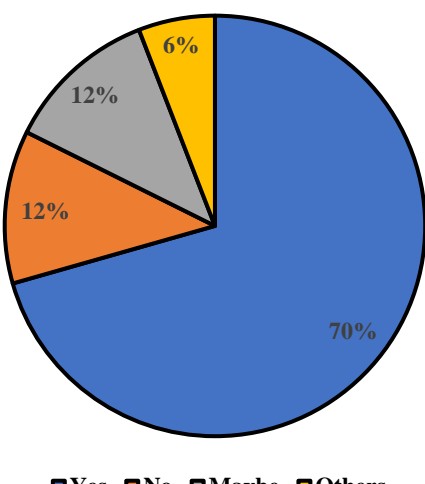

**Figure 1.** Respondents' understanding of IMO emission control and impact.

The next questionnaire about the shipping company plan is on its use as an alternative fuel and interestingly only 35.3%, 17.6% and 29.4% plan to use, not use and consider using it, respectively. Figure 2 below illustrate the respond for ship owner plan on use of alternative marine fuel.

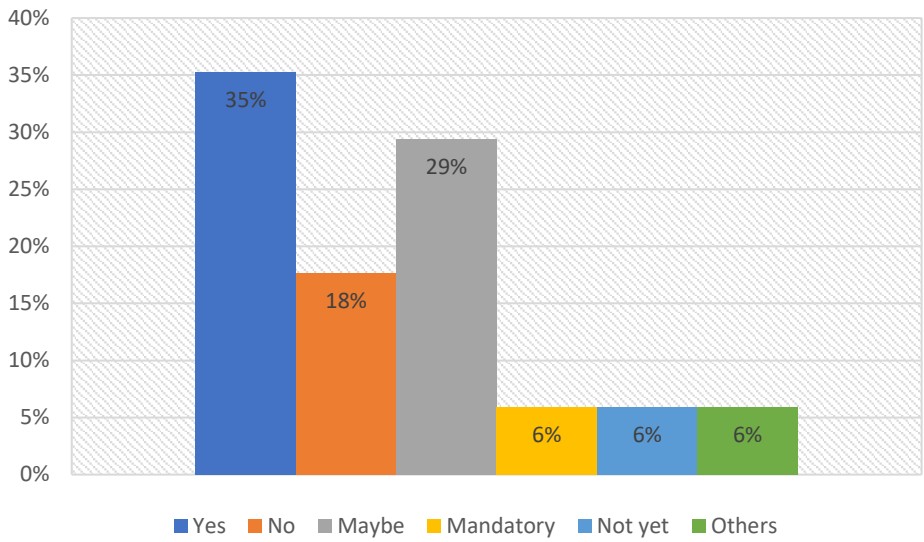

**Figure 2.** Plan for using alternative fuel.

Furthermore, approximately 45% of the respondents intend to use LNG, while 5% used other alternatives such as ethanol, methanol, and hydrogen. The remaining 14% and 9%, intend to use electricity and LPG, respectively. With respect to this question, respondents can use more than one energy alternative as planned. Figure 3 below illustrate the respond to alternate fuel for Indonesia shipping sector.

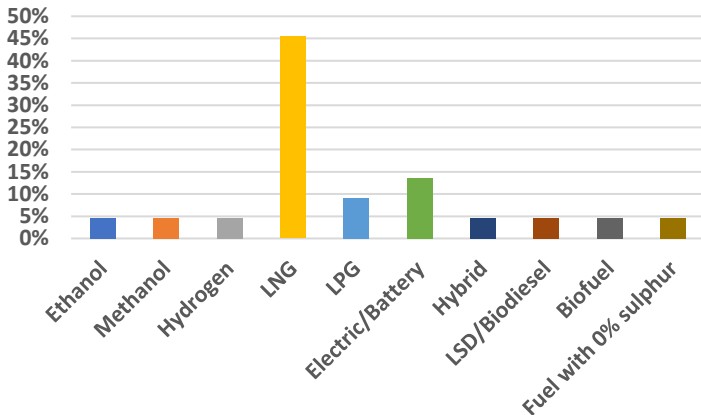

**Figure 3.** Alternative fuel for Indonesia shipping sector.

The majority of the shipping companies stated that the selection of alternative fuel was based on the considerations of low price investment (30%), government regulation or authorization (25%), advanced technology (25%), energy content (10%) and resource (10%). From this mapping, most of the companies tend to consider low fuel prices, which is the primary selection factor, followed by advanced technology. Figure 4 below illustrate the reason for select the alternative fuel.

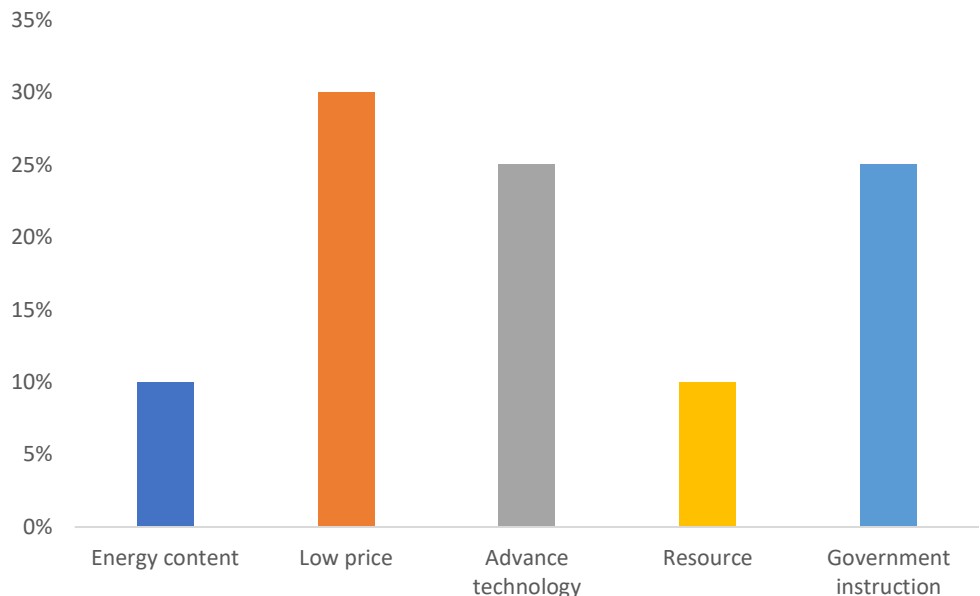

**Figure 4.** Reason for selecting alternative fuel.

Based on the earlier mentioned survey, several factors need to be taken into account by Indonesian vessel owners when investing in alternative energy. The most challenging factors to be considered are freight rate versus investment. Currently, some owners use conventional fossil fuels, such as MGO and HFO in their comfort zone. However, when IMO strengthened its position to reduce emissions from vessels, it was supported by environmental energy, which is the main objective, alongside domestic and international trade. The study of LNG-fueled vessel investment in recent years is increasing, although none analyzed the capability of Indonesian vessel owners to invest in this alternative energy. Therefore, this present study focuses on the challenges that vessel owners face in the country, specifically in understanding the investment strategy concerning the use of LNG as a marine fuel. It seeks to economically analyze this strategy by considering the potential retrofit for owners' existing fleets.

## 2. Literature Review

### 2.1. LNG Technology

The choice of alternative fuel by vessel owners is mainly driven by investment costs and advanced technology. Elkafas et al. stated that both natural gas and hydrogen are already used [16]. However, compared to natural gas, hydrogen has safety issues. The advanced technology depends on the availability of a bunker and related infrastructure. In 2019, DNV identified some alternative fuels used in shipping companies, such as LNG, LPG, methanol, biofuel and hydrogen. LNG is the most popular and promising alternative fuel because its technology is developed correctly [17]. It has been developed through significant innovation, hence, its ability to reduce the high content of fuel emissions. They also stated that the capability to reduce sulfur and nitrogen levels is due to the use of marine fuel, such as LNG, in a diesel engine [18]. Clean and renewable energies are ideal, although, in practice, LNG is usually selected by owners [19]. LNG is categorized as the leading alternative fuel, followed by methanol and biofuel [4].

LNG technology on board vessels depends on the fuel gas supply system (FGSS). Wang et al. stated that the fuel tank needs to be kept in the liquid phase at −163 °C. Furthermore, it is designed to supply gas to dual-fuel engines under the required temperature and pressure. It also needs to avoid being over-pressurized due to its ability to improve fuel efficiency [20]. Most vessel technology uses dual fuel systems, while the boil-off gas produced in the LNG tank is used for steam turbines [21]. In 2022, the Maritime Executive stated that the retrofit concept reduces the cost of LNG conversion operations [22].

### 2.2. Investment Analysis Outlook

Generally, vessel owners need a reference for their investment because they usually encounter difficulties, such as changing the current fuel to an alternative one that is environmentally friendly. Some previous studies stated that as a marine fuel, LNG would positively impact the future; its technologies are bound to pay off in a matter of years (DNV-GL, 2015). Some methods can help owners adopt an ideal investment strategy, for example, the cash flow. The uncertain price of LNG is also a huge drawback for transitioning to alternative fuel. Chen et al. stated that no international market is currently dealing with natural gas. Furthermore, the common economic analysis approach that considers time value, namely present, final, and annual worth methods, is employed in selecting these alternatives [14]. Some literature stated that most shipping investment evaluations use Real Options Analysis (ROA). ROA is used because it incorporates the uncertain prices of both LNG and conventional fossil fuels [14]. Previous studies stated that the shipping investment decision is based on the relation vessel between the current freight and trigger rates from ROA and Net Present Value (NPV). Kou et al. stated that it impacts the mean freight rate [23]. Figure 5 illustrates the challenges Indonesian vessel owners face regarding the regulation requiring them to comply with emission control. Another economic assessment method is using life cycle cost assessment (LCCA) which this method used to investigate the total cost including the sum of investment, maintenance and operations costs [24]. LCCA is a method for analyzing cost throughout the lifecycle of a product or service and it is a preferred method for the decision-making process. The LCCA method has been demonstrated to be effective when it is used for assessing the yacht cost model [25]. In the shipping industry, it is difficult to assess fuel prices for certain periods and the result from one study showed that the sensitivity of lifecycle cost for uncertain fuel prices can be observed [26]. In terms of LNG fuel options, a study from Alvestad which compares MGO, LNG and scrubbers has concluded that for new build vessels, LNG fuel might be the most economical marine fuel alternative [27].

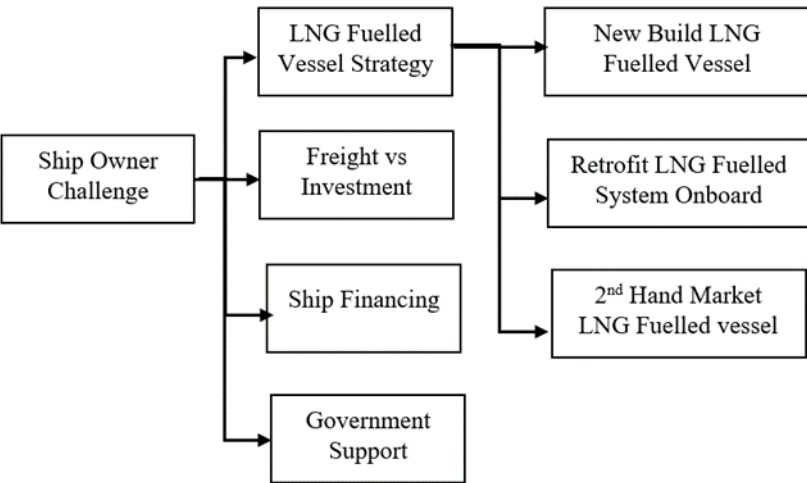

**Figure 5.** Indonesian vessel owners challenging condition.

As mentioned earlier, vessel owners need to consider the capital cost of the initial investment whenever they want to use alternative fuels such as LNG. The owners were exposed to three options, namely building new LNG-fueled vessels, retrofitting and purchasing from the second-hand market. New build and second-hand purchase markets depend on sales, while their characteristics are centered on the vessel type [28]. According to Rivieramm News, DNV reported that 240 LNG-fueled vessels were ordered in 2021, consisting of the container ship, tanker and bulk carrier sectors. Snyder further stated that based on DNV data, 251 LNG-fueled vessels are presently in operation globally, and 403 fleets are under construction [29]. This implies that the development of LNG-fueled vessels has progressed significantly. Some studies were carried out to analyze the investment in fuel transition. [20] Wang et al. calculated the low-cost analysis (LCC) for boil-off gas management and discovered that the universal solution is not applicable in all situations. It was further stated that the fuel gas supply system depends on the vessel's scale, operation, and LNG fuel price [20]. Yoo conducted an economic assessment of LNG as marine fuel for $CO_2$ carriers and compared it to MGO. It was found that LNG is more cost-effective compared to MGO. He also used the discount rate, and the project lifetime functions to calculate the annual cost index on LNG and MGO [30]. According to studies on Discount Cash Flow Method (DCFM) LNG fuel container vessels with low-speed diesel attract economic investment compared to the Tier III complied oil-fueled container vessels [26].

### 3. Methodology

*3.1. Selection for Vessel*

Vessel owners who invested in fuel transition are demanding to know when they can benefit from vessels in the market. The container vessel is extremely important in the Indonesian shipping industry and has an impact on emission control regulation. Other factors that need to be considered during selection are tankers, offshore supply vessels, tug boats, and fuels paid for or provided by the charterer. This restricts the vessel owners' flexibility to change to another alternative fuel. This study selected a container vessel with a capacity of 600 TEU because it was considered suitable and the capacity size is commonly available in the Indonesian shipping market compared to other container capacities. The container vessel with 600 TEU capacity is also the feeder size container that plays an important role in short sea shipping within Indonesia and the nearest regional countries such as Singapore and Malaysia. Furthermore, it has a company schedule and voyage, which simply means that assuming owners change to LNG, the maintenance program can be predicted and managed quickly. For this analysis, the vessel route is from the Port of Tanjung Priok, Indonesia, to the Port of Singapore, with a distance and economical speed of approximately 591 nm and 11 knots, respectively.

*3.2. Vessel Design*

In this study, the existing container vessel was used to carry out certain analyses. This is intended to provide an overview of the vessel owners' perspective on the fuel transition strategy. Assuming this is not a new build vessel, the ideal methodology that needs to be adopted is retrofit. The availability of technology reduces the cost of the vessel and improves efficiency. Furthermore, it was stated that zero-emission fuel impacts the already-built vessel [31]. It provides retrofit, which most vessel owners usually consider. The 600 TEU container vessel serves as a retrofit to dual fuel. According to the Retrofit Series (2020), the three vessels subjected to retrofit have significant potential savings, such as lube oil cleaning and other attributes that are often overlooked [32]. This includes potential savings from machine learning. Some other studies carried out on a mini-cape size bulk carrier stated that the payback period for LNG-fueled vessel retrofit is 4.5 years compared to a 0.5% compliant fuel vessel [33]. A retrofit vessel that uses LNG fuel is an attractive option to meet the new regulation. Another study stated that Hapag-Lloyd investigated a 15,000 TEU Sajir retrofitted for LNG fuel. This concept has LNG cylinders contained in open frames with 40-foot containers. The venting system and LNG piping, including the fire-fighting technique, are integrated into the container cell guide structures handling the gas adjacent to the storage. It feeds the low and high-pressure fuel gas system to the current four-stroke dual-fuel engines [22]. In this study, the 600 TEU container vessel has a similar concept with retrofit, as stated by previous studies on 15,000 TEU Sarji by Hapag-Lloyd. Wang et al. designed a three-configuration fuel gas supply system, and as mentioned earlier, FGSS is a critical factor in the LNG fuel system. The three configurations of FGSS are GCU, AE, and reliquefaction schemes with the combustion of boil-off gas-by-gas combustion unit (GCU), supply boil-off gas using auxiliary engine (AE) and reliquefaction boil-off gas by reverse Brayton cycle (RBC) system, respectively. In line with a previous study [20], this present study selected a suitable retrofit configuration for a 600 TEU container vessel dependent on a GCU scheme because the system is reliable, simple, and compact. Figure 6 illustrates the configured FGSS with boil-off gas handled by GCU. The configured FGSS is adapted from Wang et al.'s scheme [19], and the LNG Tank is fitted into a deck with a similar arrangement as a refrigerator container tank. It uses the plug-in system on the LNG tank and container cell, thereby reducing the cost of the conversion vessel [22].

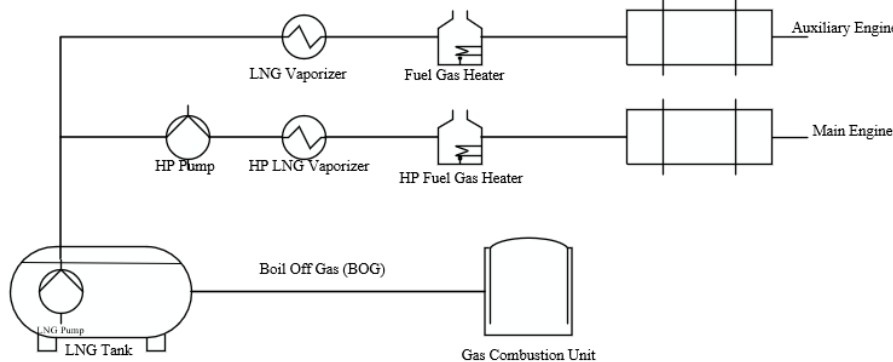

**Figure 6.** Configuration of FGSS with BOG handled by GCU.

The retrofitted designed vessel has employed a promising strategy to avoid uncertainty. They defined the retrofit cost using a Pareto-optimal solution, and interestingly, it depends on different alternative fuel types. The retrofit cost was calculated by analyzing certain aspects, namely machinery, tank, piping, shipyard and lost income. Figure 6 shows an illustration formulated by Lagemann et al. [34] as a reference. Another study that proposed the use of the calibrated method for the fuel substitution ratio, economy and particulate matter emission proved brake-specific consumption for the dual fuel model is higher than the diesel [35]. The generated boil-off gas tends to have certain advantages, such as energy efficiency [36]. The calculated lost income and the time needed during retrofit at

the shipyard are perceived as a challenge to Indonesian vessel owners because it requires opportunity costs to compensate for the time lost.

### 3.3. Maintenance and Crew Cost

This study defined and considered three maintenance scenario assumptions. These consisted of high, medium, and low scenarios dependent on the Moore Maritime Index. For a high scenario, the assumption of all maintenance and crew costs is increased by 10%. Meanwhile, for the medium scenario, there is no difference between existing and retrofit vessels, and for the low medium, the lubricating oil and spare parts are reduced by 50%, as opposed to maintenance costs by 33% less than the initial. The maintenance and crew costs are the most significant operational expenditure, besides fuel oil prices.

Table 1 shows the information of these three scenarios with operational cost in accordance with Moore Maritime Index 2021 [37] on an average level.

**Table 1.** Maintenance and crew cost scenario.

|  | High Scenario | Medium Scenario | Low Scenario |
|---|---|---|---|
| LNG price | USD 1100/MTon | USD 1100/MTon | USD 1100/MTon |
| Maintenance cost and crewing | 10% increase | No difference between existing and retrofitted vessels | Lubricating Oil reduce by 50%, Spare parts reduce by 50%, and maintenance becomes 33% lesser than the initial cost |
| Operational cost based on Moore Maritime Index | Average | Average | Average |

### 3.4. LNG Fuel Prices

LNG fuel prices are the dominant factor in determining the economic analysis of its transition. The fuel cost depicts approximately 60 to 80% of the total operating cost, while the rising oil price poses a huge challenge [38]. From the beginning of the fuel transition plan, the vessel design is not altered since the vessel will be retrofitted, and the changing prices tend to impact the LNG fuel system [26]. The increasing LNG fuel cost affects the overall system as well. Further, Wang et al. stated that the preference for appropriate FGSS configuration depends on the LNG price [20]. Some other studies stated that the LNG investment option depends on three parameters. These include the price differentials between LNG and conventional fossil fuels, new build LNG fueled vessels compared to the conventional type that entails burning traditional maritime fuels, and the shared operations within ECAs. They also observed the cost change in different bunker locations, such as Japan [14]. The major source of LNG fuel prices referenced in the market is Henry Hub for the east coast US and TTG or NBP for North West Europe and Asian markets. Japanese prices are perceived as an option. Figure 7 shows the illustration of the marine fuel price differential [39], which is cheaper based on a negative differential.

Lagemann et al. also described the fuel prices for some alternative fuels within a certain period [34]. This group of fuel types was sorted based on prices and divided into fossil, bio, and e-fuel.

### 3.5. Flow Analysis

Another challenge encountered is that the Indonesian government has yet to implement green environmental fuel regulations to support vessel owners to change from conventional to alternative fuel. The government must provide some incentives to attract these individuals to use alternative fuels such as LNG. Figure 8 illustrates Indonesian vessel owners' challenging situation before changing their fuel management to an alternative type, such as LNG.

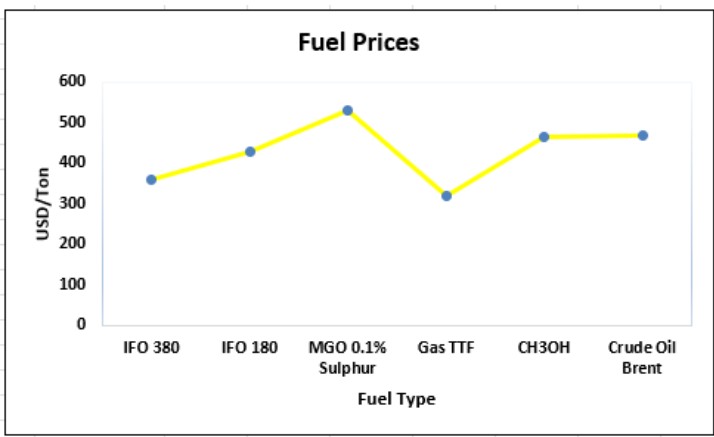

**Figure 7.** Marine fuel price differentials.

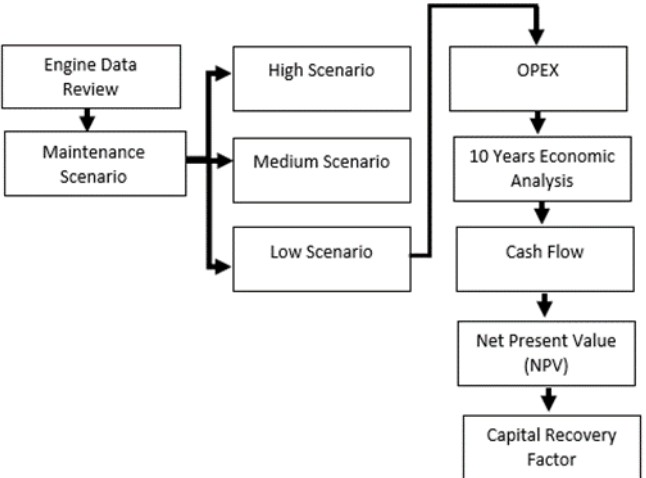

**Figure 8.** Flow Diagram for data analysis.

The data were analyzed using the lower scenario, assuming that the maintenance cost and spare parts were reduced due to LNG usage. This study employed some processes to obtain the cost recovery factor for investing in LNG fuel existing 600 TEU vessels. The first step for the analysis is gaining engine information from vessel owners. The expected data are engine power, specific fuel consumption, and type. The next step is to select the maintenance and crew cost scenarios. In addition, this study compared the high and low scenario investments. A particular study on Niigata's engine manufacturing proved customer satisfaction with gas engine series with low running cost and required maintenance at 4000 h running intervals [40]. Wartsila stated that switching to LNG as a marine fuel, whether new build or converting existing technology, will generate significant savings in fuel cost, thereby increasing profitability [41].

Operating Expenditure (OPEX) consists of the spares, lubricant, repair and maintenance. Retrofit investment for container vessel 600 TEU constitutes modifying the fuel and gas supply system. This is realized by installing the LNG tank, piping, changing the main engine and installing a gas combustion unit. As discussed earlier, vessel owners usually consider these, including vessel modification and additional equipment installation. The economic analysis requires a 10-year scheme because it is sufficient to review the potential payback from the vessel owners' view and offers future plans for vessel acquisition. Net Present Value (NPV), with respect to a 10-year investment scheme, shows differences between the present cash inflows and outflows.

Furthermore, the vessel owner requires an analyzed loan payment in a different scenario. It is calculated based on loan principal, interest, payment, and remaining amount.

Some formulas used to calculate NPV and CRF [20] are as follows:

$$\text{NPV} = \sum_{t=1}^{t} \frac{C_t}{(1+r)^t} - C_0 \tag{1}$$

where:
$C_t$ = cash flow for time (t)
$r$ = interest rate
$C_0$ = initial investment on year 0
$t$ = time

$$\text{CRF} = \frac{i(1+i)^n}{(1+i)^n - 1} \tag{2}$$

An economic analysis of the investment in trans-ocean LNG-fueled container ships, 9300 TEU sailing between Asia and Europe, showed that the LNG low-speed diesel vessels compared to oil-fueled SCR is more attractive [27]. Furthermore, Adachi et al. discovered that the NPV with a lifetime of 20 years is larger, while the refund time to payback is shorter for LNG vessels. Wang et al. also economically analyzed the lowest CAPEX for retrofit fuel gas supply systems and discovered a lower LCC on the auxiliary scheme [18]. LCC analysis allows the assessment of some shipping costs. These include acquisition (capital cost), operation or running costs, fuel consumption, operational services, maintenance, and ship disposal costs [25]. Some of the important elements of this economic assessment can be defined as follows.

### 3.5.1. CAPEX

Capital expenditure or cost is an essential element that owners consider when making investment decisions. Since the retrofit approach was employed, investing in conversion vessels has become a fundamental option. Wang et al. stated that there are three fuel gas supply systems, and the one with a gas combustion unit has a lower cost. However, this study used an FGSS with the GCU approach as the capital cost includes direct and indirect prices. Direct cost is related to purchase, installation and other related labor expenditures. The indirect costs are related to transportation, insurance, tax, construction overhead, and engineering expenditures [20]

### 3.5.2. OPEX

Operating expenditure is all the costs related to operational activities, such as maintenance and crew costs. Wang et al. stated that, unlike onshore LNG plants, the FGSS has varying fuel consumption during the voyage [20].

The total expenditure in the lifetime system for LCC includes CAPEX and OPEX costs [18]. Furthermore, when the CRF has been determined, it is multiplied by the CAPEX using the formula:

$$\text{LCC} = \text{CAPEX} \times \text{CRF} \times \text{n} + \text{OPEX} \tag{3}$$

The use of LNG fuel after conversion is perceived as an annual saving despite the different fuel price increments per remaining vessel life cycle [42]. It includes emission reduction with respect to the alternative fuels used in the vessel.

## 4. Case Study: Economic Analysis 600 TEU Fuel Transition from MFO to LNG

Based on an economic perspective, this study analyzed the existing 600 TEU container vessel transition from MFO to LNG to determine the life cycle cost (LCC) of a 10-year investment retrofit scheme. The container vessel plies from Tanjung Priok, Indonesia, to Singapore, at a distance of 591 nautical miles. Therefore, the existing vessel will have to be retrofitted to the LNG system, and the major information extracted from the technical analysis of the owners' plans to change fuel, specifically the data concerning the Specific Fuel Oil Consumption from both main and auxiliary engines. The type of fuel used is MFO and Table 1 shows the estimated price of the new build 600 TEU container vessel in the

market. PT Samudera Indonesia purchased this vessel for 8.5 million USD in 2018 from Jingjiang Nanyang Shipbuilding China [43]. This container vessel was an MFO-fueled vessel. Table 2 shows 2% inflation per year for new build vessels manufactured with FGSS installed on board culminating in two million USD. The estimated cost is based on 600 TEU newly build MFO fuel container vessel prices from 2018 from PT Samudera Indonesia and calculates 2% inflation each year.

**Table 2.** Estimated new build 600 TEU container vessel.

| New Build with MFO Fuel (MUSD) | New Build with LNG Fuel (MUSD) |
| --- | --- |
| 9,180,000 | 11,180.000 |

Furthermore, Figure 9 adapted from the Moore Index 2021 illustrates a container vessel OPEX with various sizes. There are no data for those below 1000 TEU as per the subject size vessel used in this study. For the new build, the analysis was carried out using vessel between 1000 TEU to 1999 TEU. The Moore Index was used to determine each sub-category independently. This study calculated the total OPEX expenditure, which includes maintenance and crew costs using the Moore Index as a reference.

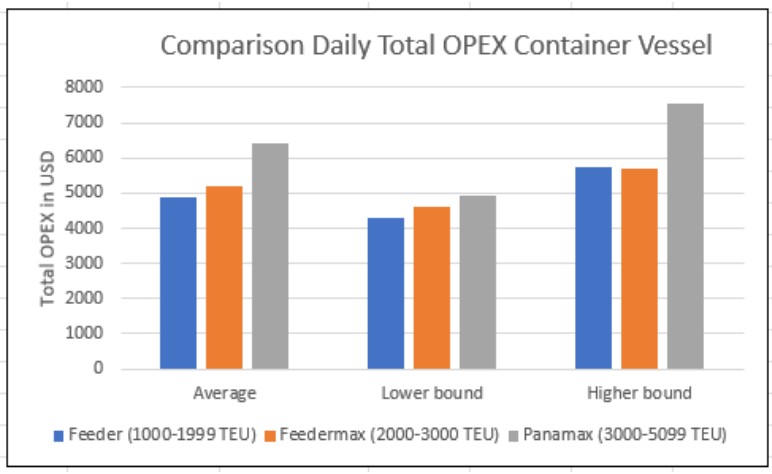

**Figure 9.** OPEX cost for various container vessels.

### 4.1. Cost Assessment for Retrofitting 600 TEU Container Vessel

This analysis was centered on the assumption of retrofit cost in 2022, which encompasses main, and auxiliary engines, fuel gas supply system and installation. The conversion cost is USD 200 to USD 340 per HP, based on the upper bound assumption [39] Furthermore, this retrofit has an estimated cost of USD 3,600,000. Figure 10 is adapted from Lagemann et al. [26] and illustrates retrofit costs for various fuel types.

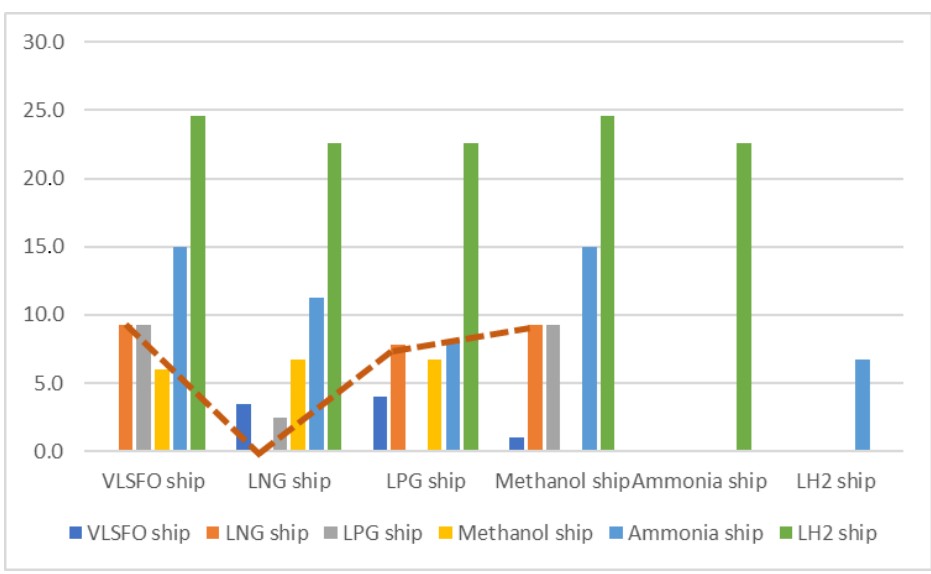

**Figure 10.** Retrofit cost various types of fuel vessel.

Banawan et al. stated that using gas as fuel reduces the deposit of organic material in the combustion chamber [43]. The reduction in hydrocarbons and other particles from the fuel affects its mass deposit. Some studies carried out on LNG as a marine fuel stated that the maintenance cost is reduced because a small amount of lubricating oils is applied on the spare part compared to the engine system using MFO or HFO. By reducing emissions, the annual expenditure determines the entire cost of natural gas applied on the main fuel onboard, including the capital expenses due to conversion [25].

Based on OPEX per year, the equivalent loading and offloading per year is 312 days. This is because the sailing duration from Jakarta to Singapore lasts for approximately three days. Based on an interview session held with one of the owners, the loading and offloading duration is usually two days for one trip, with the assumption that the in-container at the terminal is 60 containers/day.

Table 3 shows the engine information that it used for investment calculation which consist of data regarding power, number of unit, specific fuel oil consumption and type of fuel.

**Table 3.** Engine information.

|  | Main Engine | Auxiliary Engine |
|---|---|---|
| Power | 2500 kW | 450 kW |
| Unit | 1 | 1 |
| SFOC | 183 | 213 g/kWh |
| Fuel | MFO | MFO |

Table 4 shows that the total OPEX/year after retrofit is usually within the range of minimum, average and maximum variables. All tend to be reduced according to the acquired data. This Table 4 ilustrate operation cost of spares, repair, maintenance, and lubricant based on the Moore Maritime Index [34]. The total OPEX per year was calculated using three variables, namely minimum, average and maximum.

**Table 4.** OPEX Calculation per year.

|  | Minimum USD/Day | Average USD/Day | Maximum USD/Day |
|---|---|---|---|
| Spares | 344 | 407 | 407 |
| Repair and maintenance | 154 | 309 | 442 |
| Lubricant | 2301 | 2488 | 2738 |
| Total OPEX/year after retrofit | Million USD/day 0.3146 | Million USD/day 0.3679 | Million USD/day 0.4143 |

The fuel consumption of a 600 TEU container vessel that uses MFO is 6296.66 tons per year based on daily SFOC multiplied by 312 days of operation. For the calculation, the yearly consumption of LNG, based on the heating value of diesel oil, is 4958.2 tons per year or 228,871.27 MMBtu. Table 5 shows yearly fuel consumption LNG and MGO.

**Table 5.** Yearly fuel consumption LNG and MGO.

| Yearly Fuel Consumption LNG and MGO | |
|---|---|
| MGO (Ton/Year) | LNG (MMBtu/Year) |
| 6296.66 | 228,871.27 |

The annual pilot diesel fuel at the terminal is approximately 10% of the total MFO consumption per year or 629.67 tons [25]. The annual fuel price for LNG is 6.27 million USD compared to MFO, which is 8.13 million USD. Therefore, there is a difference of 1.86 million USD between LNG and MFO. It simply implies that the use of LNG is more economical compared to MFO. Supposing the annual OPEX is calculated yearly based on Moore Maritime Index, 2.228 million USD will be realized, meaning LNG is more economical.

Based on an interview with one of the vessel owners, the economic analysis for a 10-year scheme is shown in Table 6.

**Table 6.** Period 10-Year Scheme for Economic Analysis.

|  |  | Data |
|---|---|---|
| Cost Component | Unit | Total Cost |
| CAPEX | USD | 3,617,624 |
| OPEX Change | USD/Year | 367,860 |
| Project Duration | Years | 10 |
| Annual Depreciation | USD/Year | 180,881 |
| Disposal/Salvage Value | USD | 1,808,812 |
| Tax Cost | %/Year | 22% |
| Inflation Rate | %/Year | 4.5% |

CAPEX was calculated using an approach based on a literature review, and USD 3,617,624 was realized. OPEX data were not given, and the interviewee only mentioned the profit per container, which is 10 USD. It simply implies that only 80% of the container vessel space is occupied by a total of 600 TEU. Assuming the vessel uses LNG as fuel, only 480 TEU container is conveyed on every single trip from Jakarta to Singapore. The total number of trips from Jakarta to Singapore is 48 trips per year. Target BEP (Break Even Point) for this analysis is 10 years with equity from the company of approximately 40% and 60% loan. Table 6 shows that the CAPEX obtained is USD 3,617,624, while the OPEX realized for 3 years is USD 183,900. Meanwhile, 60% of CAPEX and OPEX amounted to USD 2,280,933. This study used a bank interest of −8%, and in accordance with further calculations, the loan principal is USD 228,093. This loan repayment needs to be taken into consideration during LCC calculation under different vessel financing model scenarios created by each vessel owner.

Figure 11 illustrates a loan payment scheme for 10 years, as follows.

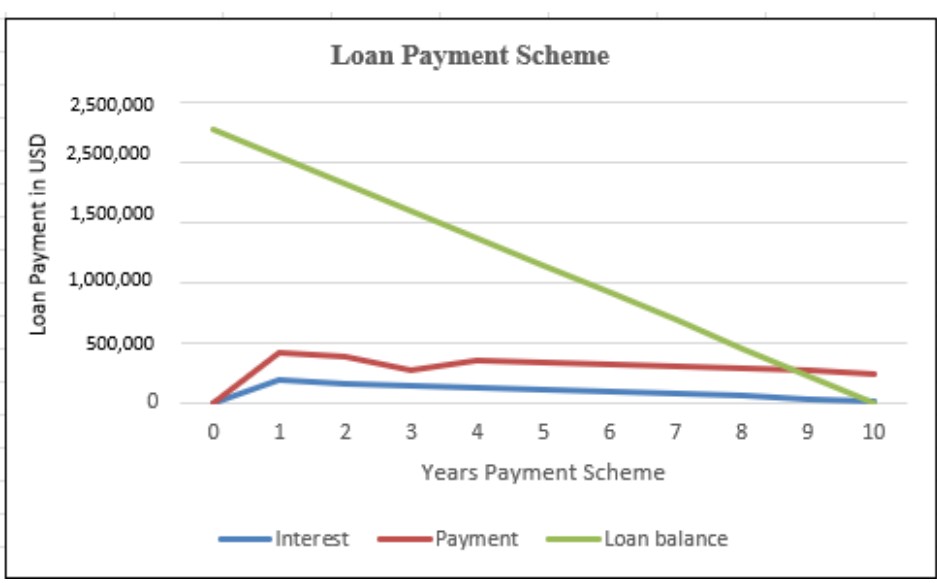

**Figure 11.** Loan payment illustration.

Cash flow is an important indicator of economic feasibility concerning the investment in the retrofit 600 TEU containership. Figure 12 shows an illustration of cash flow for 10 years. Vessel owners will encounter challenges in cash flow until the second year, and from the third, there is bound to be positive cash flow.

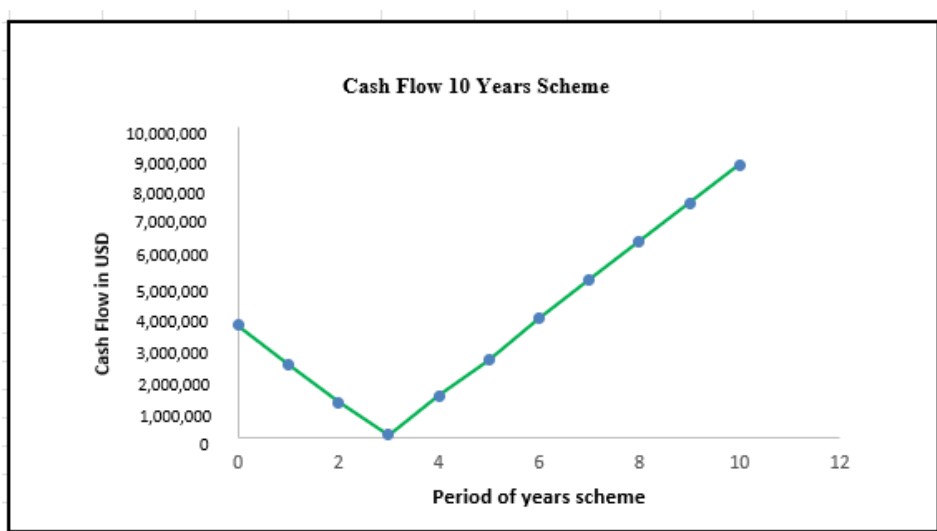

**Figure 12.** Cash flow for 10 years scheme.

For 10 years, the NPV with four Discount Factor (DF) variables tends to provide initial payback, which is used to calculate the Capital Recovery Factor on the investment scenario. Assuming the initial investment has a negative value, it is considered a capital expenditure. However, this scenario's initial cost (-) is USD 3,617,624.

Figure 13 shows the calculated NPV using various discount factors 25%, 30%, 35% and 40% and a positive NPV was realized during the 10 year scheme on the investment. The same interest rate for 10 years of investment was used for the calculation.

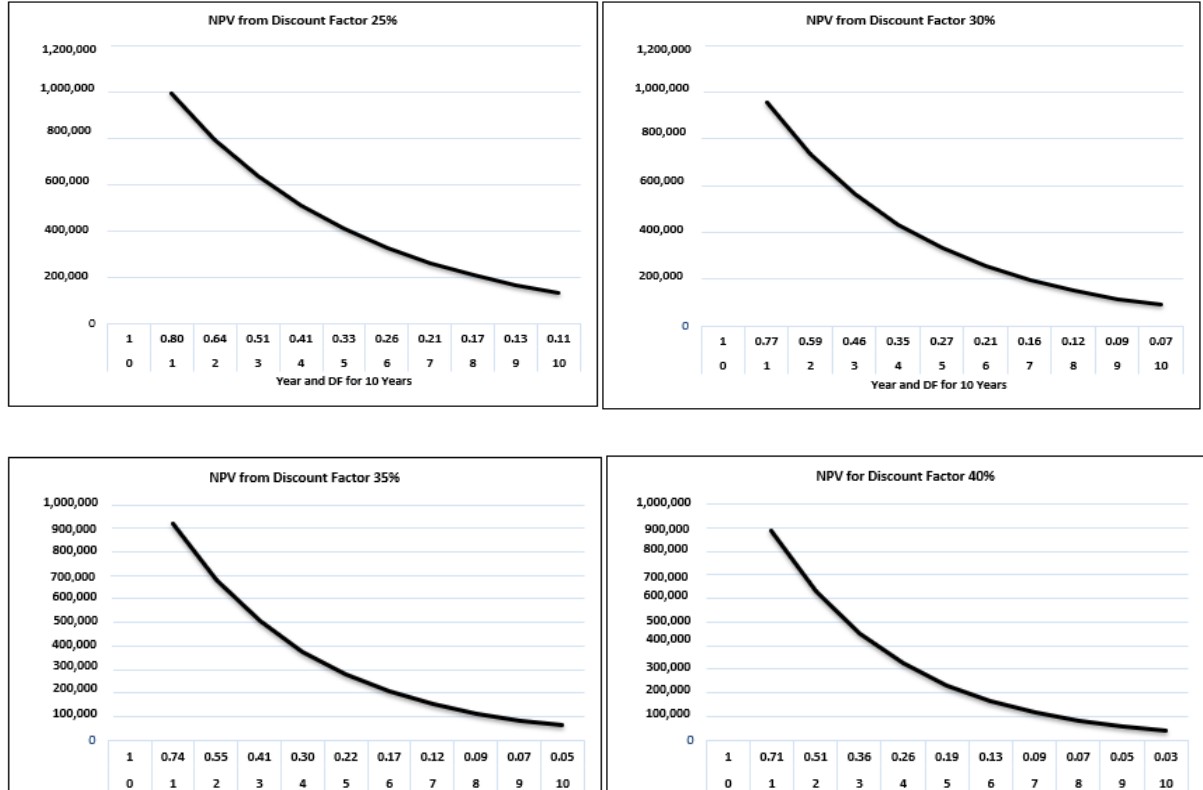

**Figure 13.** Net Present Value (NPV) for 10 Years Investment with various discount factor.

Furthermore, Figure 14 shows the Capital Recovery with three interest types such as 5%, 10%, and 15%. The CRF factor for an interest rate of 0.1 is 0.16, which was realized using the formula [2].

### 4.2. Economic Analysis

In accordance with the data acquired from the retrofit vessel, the new build 600 TEU that uses MFO and LNG fuels are shown in Figure 15. It is evident that the retrofit vessel tends to have a good competitive value compared to the MFO and LNG fuel used in the new build vessel. For the new build LNG fueled vessel, assume the OPEX is similar to the retrofit; the design will use an FGSS gas combustion unit system. The LCC of three 600 TEU container types is shown in Figure 15.

Figure 15 shows that the retrofit vessel with CAPEX on the FGSS only provides low cost with respect to the economic analysis. New build container vessels with LNG fuel consider the initial capital cost compared to the one that uses MFO. However, the OPEX cost on LNG fuel vessels continues to decrease while the vessel experiences low cost compared to the one that uses MFO. The future trend is cost-efficient for LNG fuel vessels.

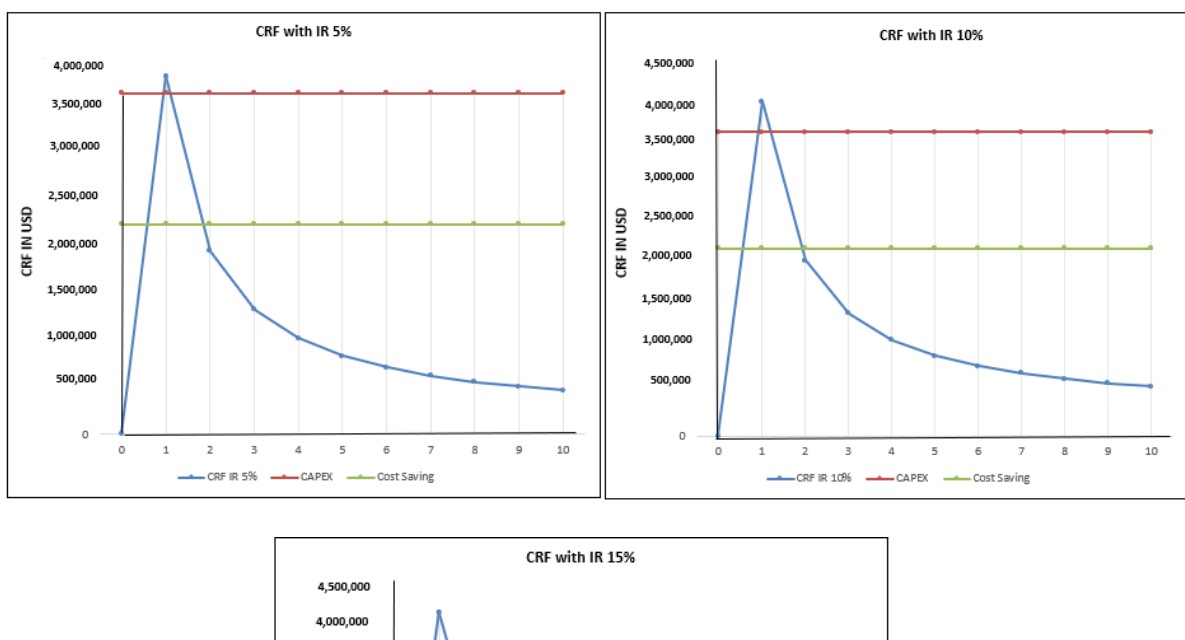

**Figure 14.** Capital Recovery Factor for 600 TEU with various interest.

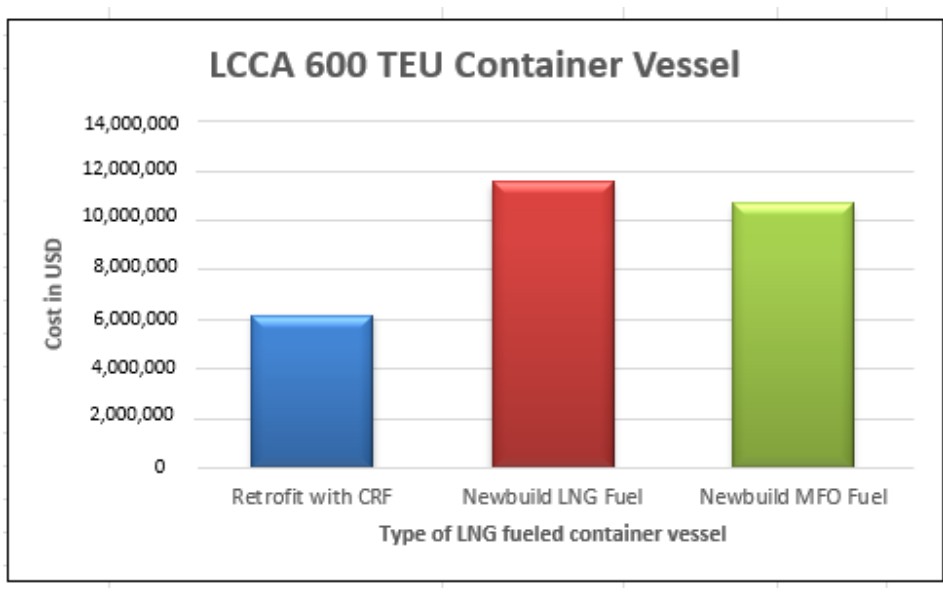

**Figure 15.** LCCA 600 TEU container.

*4.3. Sensitivity Analysis*

The sensitivity analysis shows the life cycle investment using the retrofit method for the transition process to LNG fuel, alongside some factors that influence the evaluation.

### 4.3.1. Selection of Technology

The selection of technology has an important impact on retrofit. FGSS with GCU provides low-cost investment while the implemented advanced system depends on the sailing time of the 600 TEU container. Boil-off gas is one of the factors irrespective of whether or not a longer sailing time would have an impact on its loss.

### 4.3.2. LNG Prices

LNG prices are also a critical analytical factor. The increasing LNG prices also have an impact on the overall LCC analysis. However, its uncertainty is one factor that needs to be considered in the present analysis.

## 5. Conclusions and Recommendations

The prospect of LNG as a marine fuel in Indonesia is growing because LNG is the most advanced technology for alternative marine fuel compared to other alternatives such as hydrogen, methanol and LPG. Furthermore, in terms of investment, LNG has shown good cost efficiency in long-run operations. In vessel design, the ship owner has the option to choose to retrofit technology for their current fleet instead of purchasing new vessels. The life cycle analysis of the retrofit 600 TEU container showed that the retrofit will bring low operational costs for vessel owners. This is aside from the investment, which is mostly for the FGSS on board the vessel. It helps owners to know when to use a retrofit to purchase a new build during the acquisition of a vessel. The FGSS with gas combustion unit is the first option to consider during the selection of technology. However, three comparisons made between retrofit and the other two new build shows that retrofit was the recommended option; the cost of retrofit of USD 6,156,058 is lower than the other two options for LNG fueled 600 TEUs container ship's new build (USD 11,547,860) and MFO fueled 600 TEUs container ship's new build (USD 10,702,872). Other savings that shipowners can obtain from retrofit is less time in dry dock for conversion from current MFO fueled to LNG fueled. The more time that the shipowner can save will provide an opportunity cost for the vessel to return to operation and generate income. LCCA is a tool used for life cycle and low-cost analysis with respect to retrofit investment. This analysis will be affected by LNG prices, especially when the uncertain price of LNG will bring a change in analysis.

LNG is one of the advanced technologies of alternative fuel and several studies proved that it is the most reliable energy source. From the economic analysis, it was discerned that LNG as a marine fuel reduces maintenance and spare part costs. With variable interest rates, the capital recovery factor shows a decrease in payment. The maintenance cost takes significant consideration due to the usage of LNG as fuel. The 600 TEU container vessel capital recovery result served as a reference or guide to vessel owners to be committed to using green fuels such as LNG.

This study already provided information about the challenges that Indonesian vessel owners face when they want to implement green alternative fuels. Some of these challenges are centered on technology, investment and potential profit. However, this is an opportunity for vessel owners to consider the use of LNG as marine fuel due to its long-term impact on cost efficiency and operating activities.

**Author Contributions:** Conceptualization, R.B. and R.O.S.G.; methodology, R.B.; software, R.B.; validation, R.B., R.O.S.G. and S.; formal analysis, R.B.; investigation, R.B.; resources, R.B. and R.O.S.G.; data curation, R.B.; writing—original draft preparation, R.B.; writing—review and editing, R.B. and R.O.S.G.; visualization, R.B.; supervision, R.O.S.G.; project administration, R.B.; funding acquisition, R.B. All authors have read and agreed to the published version of the manuscript.

**Funding:** This research received no external funding.

**Institutional Review Board Statement:** Not applicable.

**Informed Consent Statement:** Not applicable.

**Data Availability Statement:** Not applicable.

**Conflicts of Interest:** The authors declare no conflict of interest.

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
