# Peer review of "Prospect of LNG as Marine Fuel in Indonesia: An Economic Review for a Case Study of 600 TEU Container Vessel"

_applsci, doi:10.3390/app13052760_

Round 1
Reviewer 1 Report
Major comments:
LNG, as a fossil fuel, is for sure not the fuel to achieve IMO zero emission targets by 2050 - please discuss!
About the survey - no info on companies, neither number of respondents are shown. Please, add more justification to these results.
Chapter 3.1. is completely unclear - by this approach you can select any other vessel, not only 600 TEU CS
In Table 1, the LNG price is the same for all scenarios? Is this ok?
Please add references to formulas (1) and (2) - they exist in the literature.
Eqs. (3) and (4) should be equal, but in your case they are equal only if CRFxn=1 - is that the true?
How did you get results in Table 2?
The role of Figure 10 is completely unclear here? Is it really relevant to the ship considered in this paper?
Minor (but important comments):
- English needs to ne significantly improved - for instance statement "...uses MFO is relatively 6296,66 389 tons per year..." - word "relatively" does not belong here
- Each diagram should have name of the axis - it is omitted in most cases
- Quality of some figures is very bad (for instance Figs. 7, 9, 13)
- Literature is rather weak - there are number of papers discussing alternative fuels for shipping (also from the viewpoint of LCCA) which are not discussed here - after more detailed literature survey the originality and the novelty of this paper over such works should be indicated
Author Response
Thank you for your kind input and corrections to our paper. Here is our reply or responses to the comments requested to our paper.
1. LNG, as fossil fuel, is for sure not the fuel to achieve IMO zero emission targets by 2050, please discuss!
Response: The comment noted and the sentence has been revised that LNG is one of the alternative fuels for emission reduction and not the fuel to achieve IMO zero emission targets by 2050
2. About the survey, no info on companies, the number of respondents is shown. Please add more justification to the results.
Response: There are eight companies participated randomly in the survey. An explanation of this has been included in the revised document. An explanation of this in one paragraph has been included in the revised document
3. Chapter 3.1. is completely unclear – by this approach you can select any other vessel, not only 600 TEU CS
Response: Compare to other container capacity sizes, 600 TEU is a common container size in Indonesia
4. In Table 1, the LNG price is the same for all scenarios? Is this ok?
Response: Yes, it’s ok because Table 1 provides the maintenance scenario in the same LNG prices
5. Please add references to formulas (1) and (2)- they exist in the literature
Response: Yes, already add a reference in relation to two formulas included
6. Eqs. (3) and (4) should be equal, but in your case, they are equal only if CRF xn=1 – is that true?
Response: Noted and already revise and we use the cost recovery factor only on the formula as we are analyzing CRF as a factor on LCC
7. How did you get the result in Table 2
Response: The new build price on 600 TEU containers with MFO fuel owned by PT Samudera Indonesia in 2018. The prices as a reference with an inflation rate of 2%/year. For LNG fuel, the additional 2 million is based on the cost of FGSS on the new build vessel. An explanation of this in one paragraph has been included in the revised document
8. The role of Figure 10 is completely unclear here. Is it really relevant to the ship considered in this paper?
Response: Figure 10 actually gives an illustration of the retrofit vessel cost as the study used retrofit as the design selection methodology
9. English needs to be significantly improved – for instance statement “…uses MFO is relatively 6296,66 389 tons per year…” – word “relatively” does not belong here
Response: The comment already elaborates in the revision as shown in the revised document.
10. Each diagram should have the name of the axis – it is omitted in most case
Response: Noted and revise. The diagrams have been revised to have name on the axis
11. The quality of some figures is very bad (for instance Figs 7,9,13)
Response: Noted and revise. The figures have been revised
12. Literature is rather weak – there are a number of papers discussing alternative fuels for shipping ( also from the viewpoint of LCCA ) which are not discussed here- after more detailed literature survey the originality and the novelty of this paper over such works should be indicated
Response: Well noted, and some additional literature have been included in its discussion

Reviewer 2 Report
Currently, the topic of this work is very relevant, because it allows to reduce the anthropogenic load on the environment.
1. The problems of using traditional marine fuel are not fully disclosed in the introduction. The list of references should be expanded, the paper considers an insufficient number of references. Describe traditional petroleum-based marine fuels as well as major alternative fuels. The following articles may help you:
https://doi.org/10.1016/j.fuel.2022.125291
https://doi.org/10.3390/jmse10081017
2. Figure 3. What does “Fuel with 0% sulfur” mean?
Links should be added to the sources from which the diagrams were obtained.
3. Lines 124-125. Does the diesel engine run on LNG?
Lines 124-127. Two sentences refer to the same source. Perhaps the duplicate link should be removed and left only at the end of the second sentence.
4. Line 132. The sign of degrees Celsius should be corrected.
5. Figure 9. On the "Total OPEX" diagram, the labels of the axes should be corrected.
6. Lines 444, 448. Figure 17 is not in this article. The figure number in lines 444, 448 should be corrected.
7. The conclusions to the article should be corrected. Add more specific data and dependencies from the article.
Dear Authors,
I have carefully read the manuscript submitted by you and believe that it fully corresponds to the subject of the Special Issue.
Best regards

Author Response
Dear reviewer 2, really appreciate the input and revisions to the paper. Here are the responses for all listed items:
1. The problems of using traditional marine fuel are not fully disclosed in the introduction. The list of references should be expanded, the paper considers an insufficient number of references. Describe traditional petroleum-based marine fuels as well as major alternative fuels. The following articles may help you: https://doi.org/10.1016/j.fuel.2022.125291; and https://doi.org/10.3390/jmse10081017
Answer: Noted and elaborate in the paper. In the paper becomes: “The shipping sector is an important player in the Indonesian economy because sea transportation is cost-effective. Its growth is impacted by indigenous and international regulatory bodies such as IMO. However, the current regulatory standard adopted by IMO is emission control from the vessel's exhaust. [1] Arefin et al stated that the increased demand for energy triggers the production of greenhouse gases (GHGs) in enormous quantities. GHGs are obtained from burning fossil fuels, which ultimately cause global warming. Since the implementation of emission control by IMO, several studies have been carried out on alternative fuels, and presently, various types are available in the market. Vessel operators have no choice but to select advanced alternative fuel technology as a management strategy. In terms of sulfur emission, the traditional marine fuel, its residual is influenced by component and hydrocarbon composition and the structure of asphaltenes [2]. The characteristics of both physical and chemical asphaltenes will also impact to sulfur content [3]. The major alternative marine in development in terms of technology and resources are hydrogen, LNG, methanol, and battery [16]. Sharples stated that transportation modes are significant sources of carbon emission (CO2) [4]. Air pollution containing SO, NOx, and particulate emissions significantly impacts human health”
2. Figure 3. What does “Fuel with 0% sulphur” mean? Links should be added to the sources from which the diagrams were obtained.
Answer: Fuel with 0% sulfur in the diagram is one of the alternative fuels that contained zero sulfur and it is only for giving option for the responder to in this survey. The diagram was the result of the survey conducted by the authors and not obtained from other sources. Therefore, no changes in the paper.
3. Lines 124-125. Does the diesel engine run on LNG
Answer: Yes, the diesel engine run on LNG. Noted to change the duplication. The changes in the paper: "It has been developed through significant innovation, hence, its ability to reduce the high content of fuel emissions. They also stated that the capability to reduce sulphur and nitrogen levels is due to the use of marine fuel, such as LNG, in a diesel engine [12].”
Lines 124-127. Two sentences refer to the same source. Perhaps the duplicate link should be removed and left only at the end of the second sentence.
Answer: Noted and already revise. The changes in the paper: "“LNG technology on board vessel depends on the fuel gas supply system (FGSS). Wang et al. stated that the fuel tank needs to be kept in the liquid phase at relatively -1630 C [15]”
4. Line 132. The sign of degrees Celsius should be corrected.
Answer: Done, already revise. The new diagram has been provided.
5. Figure 9. On the "Total OPEX" diagram, the labels of the axes should be corrected.
Answer: Done, already revise. The new diagram has been provided.
6. Lines 444, 448. Figure 17 is not in this article. The figure number in lines 444, and 448 should be corrected.
Answer: Noted and elaborate in the conclusion.
7. The conclusions of the article should be corrected. Add more specific data and dependencies from the article.
Answer: Noted and elaborate in the conclusion. The changes in the paper become: "
5. Conclusion and Recommendation
The prospect of LNG as marine fuel in Indonesia is potentially growing because LNG is the most advanced technology for alternative marine fuel compared to other alternatives such as hydrogen, methanol, and LPG. Furthermore, in terms of investment, LNG showed good cost efficiency in the long run operation. In vessel design, the ship owner has the option to choose the retrofit technology for their current fleet instead of purchasing new build vessels. The life cycle analysis of the retrofit 600 TEU container showed that the retrofit will bring about low operational costs for vessel owners. This is asides from the investment, which is mostly for the FGSS on board the vessel. It helps owners to know when to use a retrofit to purchase a new build during the acquisition of a vessel. The FGSS with gas combustion unit is the first option to consider during the selection of technology. However, three comparisons made between retrofit and the other two new build shows that retrofit was the recommended option such as the cost of retrofit USD 6,156,058 is lower than the other two options for LNG fuelled 600 TEUs container ship’s new build (USD 11,547,860) and MFO fuelled 600 TEUs container ship’s new build (USD 10,702,872). Another saving that shipowners can get from retrofit is less time in dry dock for conversion from current MFO fuelled to LNG fuelled. The more time that the shipowner can reduce, it will provide an opportunity cost for the vessel to return to operation and generate income. LCCA is a tool used to for life cycle and low-cost analysis in respect to retrofit investment. This analysis will highly be affected by LNG prices, especially when the uncertain price of LNG will bring the change of analysis.
LNG is one of the advanced technologies of alternative fuel and several studies proved that it is the most reliable energy source. From the economic analysis, it was discerned that LNG as marine fuel reduces maintenance and spare part costs. With variable interest rates, the capital recovery factor shows a decrease in payment. The maintenance cost takes significant consideration due to the usage of LNG as fuel. The 600 TEU container vessel capital recovery result served as a reference or guide to vessel owners to be committed to using green fuel emissions such as LNG.
This study already provided information about the challenges that Indonesian vessel owners face when they want to implement green alternative fuels. Some of these challenges are centered on technology, investment, and potential profit. However, this is an opportunity for vessel owners to consider the use of LNG as marine fuel due to its long-term impact on cost efficiency and operating activities.

Round 2
Reviewer 1 Report
The authors have seriously improved the manuscript according to my comments, and it is now nearly ready to be considered for publication. I would like to point out that in last 2-3 years there are many considerations on alternative fuels for decarbonization of not only long-distance, but also short-sea (freight and passenger fleet, fishing fleet) and even inland ships. Therefore I invite authors for deeper screening the literature (even MDPI journals have recent works on this) and to strengthen their state-of-the art and reference list.
In this way, their work will be much more self-contained and also visible to potential readers.
Author Response
Thank you for further suggestions that may provide better literature support of the paper.
In relation to the request of the reviever, stated: "
“The authors have seriously improved the manuscript according to my comments, and it is now nearly ready to be considered for publication. I would like to point out that in last 2-3 years there are many considerations on alternative fuels for decarbonization of not only long distance, but also short-sea (freight and passenger fleet, fishing fleet) and even inland ships. Therefore, I invite authors for deeper screening the literature (even MDPI journals have recent work on this) and to strengthen their state-of-the art and reference list.
In this way, their work will be much more self-contained and visible to potential readers.”
Answer:
Noted and has been elaborated in the journal. The paper becomes:
“The use of LNG as alternative marine fuel for decarbonization is implemented in various types of vessels. In South Korea, there was a study on LNG fuel application to new bulk carriers [6]. LNG as a marine fuel is not only implemented in deep-sea shipping, but it has implemented for short-sea shipping or domestic shipping. Fishing vessels are another type of vessel that increase the impact on the environment due to vessel emission, and these types of vessels have started to use alternative marine fuel. The study shows that LNG fuel is a good option for the fishing vessel to reduce environmental impact [7]. Another type of vessel that started to use LNG as a marine fuel is the Ro-ro ferry vessel. The conversion of Ro-ro vessels to LNG-fueled vessels will be technically feasible and a good option for local ship operators [8]. “
